# Measurement and spatio-temporal heterogeneity analysis of coupling coordination between development of digital economy and agricultural carbon emission performance

**Haisong Wang[1], Yuhuan Wu[1]\*, Ning Zhu[2]**

**1** Department of Economics and Trade, Hebei University of Water Resources and Electric Engineering, Cangzhou, Hebei, China, **2** Institute of Agricultural Economics and Development, Chinese Academy of Agricultural Sciences, Haidian, Beijing, China

☯ These authors contributed equally to this work.

\* wuyuhuan@hbwe.edu.cn

**Data Availability Statement:** All relevant data are within the manuscript and its Supporting information files.

## Abstract

The new development pattern has identified two key avenues for the sustained advancement of high-quality agricultural and rural development: digitalisation and low-carbon development. The measurement of the digital economy and the agricultural carbon emission performance, and their spatial and temporal heterogeneity, is a crucial step in promoting the spatial coordination and sustainable development of digitalisation and low-carbon agriculture. This paper employs the entropy value method, SBM model, and coupling coordination degree model to investigate the coupling coordination measurement and spatial-temporal heterogeneity of the performance of the digital economy and agricultural carbon emissions. The data used are provincial panel data from 2013 to 2021. The simulation results demonstrate that, between 2013 and 2021, the digital economy of all provinces exhibited varying degrees of growth, yet the development of the digital economy between provinces exhibited a more pronounced tendency to diverge. Concurrently, the agricultural carbon emission efficiency in China exhibited a fluctuating upward trend. The development of the digital economy and the efficiency of agricultural carbon emission were found to be highly coupled. Their coupling and coordination relationship showed a downward trend followed by an upward trend. In general, it is suggested that we should increase investment in digital economy infrastructure and technology, promote digital agricultural applications, strengthen policy guidance and financial support, establish a coupling coordination mechanism and strengthen farmers' digital literacy and environmental awareness.

## Introduction

As climate change and environmental issues continue to worsen, controlling and reducing carbon emissions has become a crucial global concern. In order to achieve global carbon

**Funding:** Supported by Ministry of Education, Industry-University Cooperation Collaborative Education Project (230825052507181). Funded by Science Research Project of Hebei Education Department (BJS2024097). Supported by Hebei Province Social Science Development Research Project (20230303051). The funders plays role in data collection and analysis, decision to publish, and preparation of the manuscript.

**Competing interests:** The authors have declared that no competing interests exist.

neutrality, it is crucial to control and reduce carbon emissions from agriculture, as it is one of the largest sources of carbon emissions worldwide. The United Nations Intergovernmental Panel on Climate Change (IPCC) reports that agricultural greenhouse gas emissions contribute to approximately 14% of total global greenhouse gas emissions. Consequently, reducing agricultural carbon emissions and achieving green and sustainable development in the agricultural economy is of paramount importance for the advancement of the goal of carbon neutrality [1–3]. Furthermore, the digital economy has emerged as a new engine for global economic development due to the rapid development and application of digital technology [4]. The digital economy can enhance production efficiency, optimise resource allocation, and reduce energy consumption and carbon emissions through data analysis, mining, and information technology applications [5–7]. By conducting a comprehensive analysis of the interrelationship between the digital economy and agricultural carbon emissions, it is possible to identify more environmentally friendly and efficient methods of agricultural production, explore the potential of the digital economy to assist agriculture in achieving green development, and gain a detailed understanding of the dynamic relationship between the digital economy and agricultural carbon emissions [8–10]. This will provide a scientific foundation for the formulation of more precise regional policies.

Many scholars have researched the relationship between the digital economy and agricultural carbon emissions. The impact of the development of the digital economy on the performance of agricultural carbon emissions is a complex process [11–13]. The development of the digital economy has the potential to both promote and hinder the reduction of carbon emissions in agricultural production. While it can improve agricultural production technology and mechanization, leading to a decrease in carbon emissions, it can also increase energy and resource consumption, intensifying agricultural carbon emissions and having a negative impact on overall performance [14–16]. Some researchers found that the rational development of the digital economy can significantly promote agricultural carbon emission performance. Therefore, it is necessary to balance the relationship between the development of the digital economy and the control of agricultural carbon emissions when formulating policies [17–20].

This effect is indirectly influenced by improving the level of agricultural socialization services. Traditional agriculture is primarily a small-scale, decentralized operation. It faces issues such as production intensification, standardization, mechanization, and inadequate management at a low level of scale [21–23]. Additionally, production technology and process standards are not yet perfect. The digital economy can effectively allocate agricultural resources, thereby reducing the cost of production of agricultural products and expanding the scale of agricultural production [24–26].

The level of development of the digital economy also affects the quality of agriculture in different regions. Meanwhile, the digital economy can directly contribute to enhancing green total factor productivity by improving production efficiency and developing green technologies [27, 28]. Additionally, the development of the digital economy has significant spatial spillover effects. It can promote the development of local agriculture and drive the development of agriculture in neighboring areas. The spatial spillover effect can promote regionalization and specialization of agricultural production, improving the overall efficiency of agricultural production and environmental protection [29, 30].

In conclusion, existing studies have primarily focused on the impact of the digital economy on agricultural carbon emissions, the digital economy on the high-quality development of agriculture, and the impact of agricultural green total factor productivity for qualitative analysis. This study builds upon existing research by establishing a coupled coordination degree model between the digital economy and agricultural carbon emissions. This study employs a quantitative approach to investigate the relationship between the digital economy and agricultural

carbon emissions. The findings can be utilized as a complement to existing research on the digital economy and the control of agricultural carbon emissions. Moreover, the results can serve as a foundation for further studies on the digital economy and agricultural carbon emissions control. It can be employed as a supplement to the existing studies on the digital economy and agricultural carbon emission control, and concurrently, it can furnish policy references for the advancement of "digitalization" and "low-carbonization" in agriculture.

The remainder of this article is organized as follows: the third section presents a theoretical analysis, the fourth section describes the research method, the fifth section conducts a detailed analysis of the research results, the sixth section conducts an in-depth discussion of the research results, and the last section is the conclusion and suggestions.

## Theoretical analysis

The rapid growth of the digital economy has brought about significant changes to the agricultural sector. The use of digital technology has enabled the agricultural production process to become more intelligent, precise, and efficient, thereby increasing the efficiency of agricultural production and reducing carbon emissions from the agricultural production process. Furthermore, the development of the digital economy has facilitated the optimization and upgrading of the agricultural industrial chain, promoted the process of agricultural modernisation, and has strongly supported the improvement of agricultural carbon emission performance.

Firstly, the development of the digital economy can provide intelligent technical support to agriculture, helping it to achieve accurate and efficient production and management.

Secondly, the digital economy can facilitate the implementation of more efficient and low-carbon production methods in agriculture by providing intelligent technical support and market distribution channels.

Thirdly, the development of the digital economy can facilitate the acquisition of more comprehensive and accurate data by the government, thereby enabling the formulation of more scientific and rational agricultural policies and planning.

Concurrently, there are evident contrasts in the extent of digital economy advancement and agricultural carbon emission efficiency across different regions. Some regions exhibit a high level of digital economy development and relatively high agricultural carbon emission efficiency, which can be attributed to factors such as a robust economic foundation, advanced technology, and conducive policy environment. Conversely, other regions exhibit a lower level of digital economy development and relatively lower agricultural carbon emission efficiency, which can be attributed to factors such as a weak economic foundation, outdated technology, and imperfect policy environment. This geographical disparity gives rise to pronounced spatial heterogeneity in the coordination relationship between digital economy development and agricultural carbon emission efficiency. This spatial heterogeneity gives rise to the fact that the digital economy development and agricultural carbon emission efficiency in different regions exhibit disparate characteristics and laws. It is therefore necessary to fully consider these differences in the process of policy formulation and implementation, and to take targeted measures to promote the deep integration and coordinated development of the digital economy and agriculture.

The preceding analysis has led to the formulation of the theoretical framework of this study, which is as follows(as shown in Fig 1):

## Materials and methods

### Data description

Given the inconsistency in the statistical figures of Hong Kong, Macau, and Taiwan, the research area is composed of 31 provinces, cities, autonomous regions, and municipalities

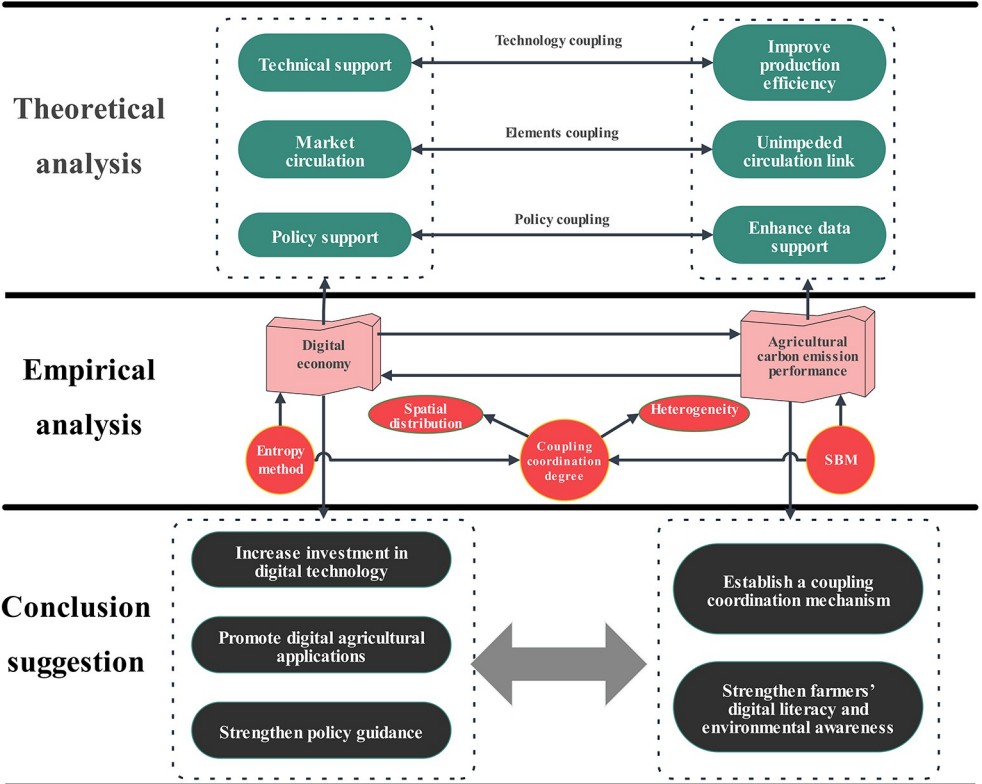

**Fig 1. Theoretical framework.**

directly under the central government, with the exception of Taiwan, Hong Kong, and Macau. The time period of this study spans from 2013 to 2021 due to data availability and statistical lag. The data presented in "Digital Economy Development Level Using the Entropy Method" is from the National Bureau of Statistics.

The gross output value of agriculture, forestry, animal husbandry, and fishery, the number of employees in agriculture, forestry, animal husbandry, and fishery, and the total sown area of crops are all data points that have been sourced from the National Bureau of Statistics.

Information regarding $CO_2$ emission factors, including those related to fertilizer, pesticides, film, irrigation, agricultural diesel, and land plowing, can be found in the United Nations Intergovernmental Panel on Climate Change (IPCC) and related websites. The detailed $CO_2$ emission coefficients are presented in Tables 1–3.

## Empirical methods and variable selection

**Entropy method.** The entropy method is a technique for determining the overall level of a system based on the degree of discreteness. Its fundamental principle involves assessing the weights of indicators by analyzing their relative changes as a whole. This allows for an accurate evaluation of the overall level. The basic steps of the entropy method are as follows:

(1) In order to eliminate the impact of differences in scale or order of magnitude on the evaluation results, it is necessary to standardize the indicators, given that there are variations

**Table 1. Carbon emissions from agricultural land use.**

| Carbon source | Metric | Data sources |
|---|---|---|
| Chemical fertilizer | 0.8956 kg(C)·$kg^{-1}$ | West et al. [34], Oak Ridge National Laboratory, USA |
| Agricultural chemical | 4.9341 kg(C)·$kg^{-1}$ | Oak Ridge National Laboratory, USA |
| Diesel fuel | 0.5927 kg(C)·$kg^{-1}$ | IPCC |
| Agro-film | 5.18 kg(C)·$kg^{-1}$ | Institute of Agricultural Resources and Ecological Environment, Nanjing Agricultural University |
| Turn the soil | 312.6kg(C)·$hm^{-2}$ | college of biological sciences, China Agricultural University |
| Irrigation | 266.48kg(C)·$hm^{-2}$ | West et al. [34] |

**Table 2. Measurement of carbon emissions from rice cultivation.**

| Carbon source | Metric | Data sources |
|---|---|---|
| Early-season Rice | 11.9418 kg(C)·$hm^2$ | Min Jisheng et al. [35] |
| Mid-season Rice | 176.072 kg(C)·$hm^2$ | Min Jisheng et al. [35] |
| Late-season Rice | 26.5980 kg(C)·$hm^2$ | Min Jisheng et al. [35] |

in their scale and order of magnitude.

$$x'_{ij} = \frac{x_j - x_{\min}}{x_{\max} - x_{\min}} \tag{1}$$

$$x'_{ij} = \frac{x_{\max} - x_j}{x_{\max} - x_{\min}} \tag{2}$$

Eq (1) use for a positive indicator and Eq (2) use for a negative indicator.
Where
$x_j$ is the value of indicator $j$
$x_{\max}$ is the maximum value of indicator $j$
$x_{\min}$ is the minimum value of indicator $j$
$x'$ is a standardized value

**Table 3. Carbon emissions from animal farming kilograms (C)/(head*year).**

| Animal | Intestinal Fermentation | | | Animal | Intestinal Fermentation | | |
|---|---|---|---|---|---|---|---|
| | $CH_4$ | $CH_4$ | $N_2O$ | | $CH_4$ | $CH_4$ | $N_2O$ |
| Cow | 370.55 | 47.74 | 101.04 | Hog | 6.82 | 27.28 | 112.96 |
| Horse | 122.76 | 11.18 | 112.97 | Goat | 34.11 | 1.16 | 26.82 |
| Donkey | 68.21 | 6.14 | 112.96 | Sheep | 34.11 | 1.02 | 26.82 |
| Mule | 68.21 | 6.14 | 112.96 | Poultry | 0 | 1.63 | 1.63 |

Source: IPCC

Note: The coefficients are for replacing $CH_4$ and $N_2O$ with standard C. The greenhouse effect of 1 ton of $CH_4$ is equivalent to 6.82 tons of C (25 tons of $CO_2$), and the greenhouse effect of 1 ton of $N_2O$ is equivalent to 81.27 tons of C (298 tons of $CO_2$).

(2) Calculation of the weight of indicator $j$ in the value of the indicator in year $i$.

$$y_{ij} = \frac{x'_{ij}}{\sum_{i=1}^{m} x'_{ij}} \, (0 \leq y_{ij} \leq 1) \tag{3}$$

Where

$i$ is 1,2,. . .,n

$j$ is 1,2,. . .,m

(3) Calculate the information entropy value for the $j$ indicator.

$$e_j = -K \sum_{i=1}^{m} y_{ij} \ln y_{ij} \tag{4}$$

Where

$K$ is a constant, $K = \frac{1}{\ln m}$

Assuming a value of $d_j$ for the information utility, and $d_j = 1 - e_j$. The weight of the $j$ indicator is:

$$W_j = \frac{d_j}{\sum_{i=1}^{m} d_j} \tag{5}$$

(4) Calculate the composite evaluation score of the sample using the following formula:

$$U = \sum_{i=1}^{n} y_{ij} \cdot w_j \times 100\% \tag{6}$$

**SBM-Undesirable model.** This paper employs the SBM-Undesirable model proposed by Cooper et al [31]. to assess the efficiency of agricultural carbon emissions in the Western region. The model effectively circumvents the bias in efficiency caused by differences in radial and angle selection, and more accurately reflects the nature of agricultural carbon emission performance. The fundamental formula of the model is:

$$\rho = \min \frac{1 - \frac{1}{m}\sum_{i=1}^{m}\frac{s_i^-}{x_{i0}}}{1 + \frac{1}{s_1 + s_2}\left(\sum_{r=1}^{S_1}\frac{s_r^g}{y_{r0}^g} + \sum_{r=1}^{S_2}\frac{s_l^b}{y_{l0}^b}\right)} \tag{7}$$

$$x_0 = X\lambda + s^-$$

$$y_0^g = Y^g\lambda - s^g$$

$$y_0^b = Y^b\lambda + s^b$$

$$\sum_{i=1}^{n}\lambda_i = 1, s^- \geq 0, s^g \geq 0, s^b \geq 0$$

Where

$m$ is the number of decision—making units

$X \in R^m$ is the input

$y^g \in R^{S_1}$ is the desired output

$y^b \in R^{S_2}$ is non-expected output

$s^-$ is the amount of input redundancy

$s^g$ is a positive output shortfall

$s^b$ is a positive output overrun

Gong Binlei's (2022) and other scholars' research on agricultural carbon emissions indicates that the output indicators are the gross output value of agriculture, forestry, animal husbandry, and fishery, and agricultural carbon emissions [32, 33]. The input indicators include labor, land, fertilizer, and agricultural machinery. The labor input is measured by the number of employees in agriculture, forestry, animal husbandry, and fishery. The land input is measured by the total sown area of crops. The machinery input is quantified by the total power of agricultural machinery. The fertilizer input is measured by the pure amount of fertilizer applied to agriculture.

With regard to the calculation of carbon emissions from agriculture, existing studies generally classify the sources of carbon emissions from agriculture into three categories. The first category encompasses carbon emissions from agricultural land use, as shown in Table 1. The second category includes carbon emissions measured from rice cultivation, as shown in Table 2. The third category encompasses carbon emissions from animal husbandry, as shown in Table 3.

**Coupled coordination degree model (CCDM).** The coupled coordination degree model provides a quantitative description of the degree of interaction and coordination among the elements within a system. It is a widely used empirical tool in the study of coupled development among various systems, including the environment, economy, social development, urbanization, agriculture, industry, transportation, and population, at different scales and in different regions. This paper employs the coupled coordination degree model to measure the coupled coordination relationship between the digital economy and agricultural carbon emission performance. The model's fundamental equation is as follows:

$$C = \sqrt{\frac{U_1 U_2}{\left(\frac{U_1 + U_2}{2}\right)^2}} = \frac{2\sqrt{U_1 U_2}}{U_1 + U_2} \tag{8}$$

$$T = \alpha U_1 + \beta U_2 \tag{9}$$

$$D = \sqrt{C \times T} \tag{10}$$

Where

$C$ is the coupling of the interaction between the digital economy and agricultural carbon performance. $D$ represents the degree of coordination of the coupling. $T$ is coordination index. The coefficients $\alpha$ and $\beta$, both taken as 0.5, are to be determined with reference to Yan Yi-Han et al. (2022) [36].

**Exploratory spatial analysis.** The Gini coefficient of location can be employed to analyze the global spatial characteristics of agricultural mechanization development. However, it is contingent upon the geographic unit settings. To comprehensively and systematically analyze the geographic agglomeration characteristics of agricultural mechanization development in China, exploratory spatial analysis is employed to examine its distribution.

Spatial correlation indicators are divided into two categories: global spatial auto-correlation indicators and local spatial auto-correlation indicators. The Moran index is employed to test global spatial auto-correlation indicators, while local indicators (LISA cluster plot) and Moran scatter plot are utilized to test local spatial auto-correlation indicators.

The calculation of the global Moran index is as follows:

$$Global \ \text{Morans}'I = \frac{\sum_{i=1}^{n} \sum_{j=1}^{n} w_{ij}(x_i - \bar{x})(x_j - \bar{x})}{S^2 \sum_{i=1}^{n} \sum_{j=1}^{n} w_{ij}} \tag{11}$$

$$S^2 = \frac{1}{n} \sum_{i=1}^{n} (x_i - \bar{x})^2$$

$$\bar{x} = \frac{1}{n} \sum_{i=1}^{n} x_i$$

Where

$x_i$ is the observed value for area $i$

$x_j$ is the observed value for area $j$

$n$ is the total number of districts

$w_{ij}$ is the element of the row $i$ and column $j$ of the spatial weights matrix

The calculation of the local moran index is as follows:

$$Local \ \text{Morans}'I_i = \frac{(x_i - \bar{x})}{S^2} \sum_{j=1}^{n} w_{ij}\left(x_j - \bar{x}\right) \tag{12}$$

## Results

### Results of the entropy method in the digital economy

This paper presents a construction of a digital economy measurement index system based on the methods of Wang, Zhong, and Peibei Shi(2021)(as shown in Table 4). The indicator system is constructed from four dimensions: digital infrastructure, digital industry integration, digital network application, and digital environment creation.

This paper selects four secondary indicators in the dimension of digital infrastructure: Internet broadband access ports, the number of web pages, the number of domain names, and the length of fiber-optic cable lines.

This paper selects six secondary indicators in the dimension of digital industry integration. These are e-commerce sales, e-commerce purchases, the percentage of enterprises with e-commerce trading activities, software sales, total telecommunications business, and employment in urban units of the information transmission, computer services, and software industry (10,000 persons).

This paper selects four secondary indicators in the dimension of digital network application. These are: all technology market turnover (in million CNY), cell phone penetration rate (in units per 100 people), digital TV penetration rate (as a percentage), and digital financial inclusion index.

This paper selects three secondary indicators in the dimension of digital environment creation: research and development (R&D) expenditures of industrial enterprises above a large scale (10,000 CNY), full-time equivalents of R&D personnel of industrial enterprises above a large scale, and the number of R&D projects of industrial enterprises above a large scale.

Table 5 presents the findings of the study, which examine the growth of the digital economy in Chinese provinces. The results indicate that the development of the digital economy varies across provinces. Only seven provinces (Beijing, Shanghai, Jiangsu, Zhejiang, Fujian,

**Table 4. Variable selection for digital economy development level using the entropy method.**

| Dimensions | Indicators | Value of indicator |
|---|---|---|
| Digital infrastructure | Internet broadband access ports(ten thousand)A1 | positive |
| | the number of web pages(ten thousand)A2 | positive |
| | the number of domain names(ten thousand)A3 | positive |
| | the length of fiber-optic cable lines(km)A4 | positive |
| Digital Industry Convergence | e-commerce sales(Billions of CNY)B1 | positive |
| | e-commerce purchases(Billions of CNY)B2 | positive |
| | percentage of enterprises with e-commerce trading activities(%)B3 | positive |
| | software sales(ten thousand CNY)B4 | positive |
| | total telecommunications business(ten thousand CNY)B5 | positive |
| | employment in urban units of the information transmission, computer services and software industry (ten thousand people)B6 | positive |
| Digital network applications | all technology market turnover (in million CNY) C1 | positive |
| | cell phone penetration rate (in units per 100 people) C2 | positive |
| | digital TV penetration rate (as a percentage)(%)C3 | positive |
| | digital financial inclusion indexC4 | positive |
| Capability for digital innovation | R&D expenditures of industrial enterprises above large scale (ten thousand CNY) D1 | positive |
| | full-time equivalents of R&D personnel of industrial enterprises above large scale(person-years)D2 | positive |
| | the number of R&D projects of industrial enterprises above large scale(Number of Projects) D3 | positive |

Data Source: the National Bureau of Statistics

Shandong, and Guangdong) have exhibited a level of digital economy development that exceeds the national average from 2013 to 2021. This indicates that the provincial digital economy has exhibited uneven development.

The varying degrees of growth in provincial digital economies and their uneven development can be attributed to three main factors. Firstly, the gradual maturation of new-generation information technology has facilitated the in-depth development of informatization, providing a solid foundation for the rise of the digital economy. Secondly, the presence of optimal digital infrastructure has also played a significant role. In recent years, China has made significant efforts to develop digital infrastructure, including 5G communication networks and high-speed broadband networks. Mobile phones and mobile internet have become widely accessible, leading to a shift in economic activities towards online platforms. This has laid a strong foundation for the growth of the digital economy. Thirdly, many provinces have issued plans to develop the digital economy and increased policy support for it. For example, Beijing, Shenzhen, Hebei, Henan, Jiangxi, and numerous other provinces and cities have implemented policies designed to stimulate and facilitate the growth of the digital economy.

With regard to the spatial distribution of China's digital economy, it can be observed that there is a general decrease in development from the coast to the inland. The core regions, including Beijing-Tianjin-Hebei, Yangtze River Delta, and Pearl River Delta, have been identified as having outstanding advantages and are expanding to the surrounding areas. In contrast, the Western region has been found to have a lower level of digital economy development, which is mainly attributed to its relatively weak digital foundation. The country's ranking in terms of the number of domain names, density of fiber optic cable routes, and density of cell phone base stations is among the lowest in the world, which has a limiting effect on the development of its digital economy.

**Table 5. Comprehensive development level of digital economy by province, 2013–2021.**

| Province | 2013 | 2014 | 2015 | 2016 | 2017 | 2018 | 2019 | 2020 | 2021 |
|---|---|---|---|---|---|---|---|---|---|
| Beijing | 0.6196 | 0.6585 | 0.6852 | 0.6609 | 0.6420 | 0.4760 | 0.6393 | 0.6450 | 0.6696 |
| Tianjin | 0.1487 | 0.1577 | 0.1520 | 0.1375 | 0.1177 | 0.0798 | 0.1138 | 0.1181 | 0.1148 |
| Hebei | 0.1448 | 0.1496 | 0.1416 | 0.1491 | 0.1547 | 0.1123 | 0.1488 | 0.1534 | 0.1550 |
| Shanxi | 0.0807 | 0.0816 | 0.0718 | 0.0744 | 0.0719 | 0.0587 | 0.0742 | 0.0753 | 0.0769 |
| Neimenggu | 0.0612 | 0.0590 | 0.0662 | 0.0676 | 0.0633 | 0.0422 | 0.0582 | 0.0612 | 0.0648 |
| Liaoning | 0.1946 | 0.1951 | 0.1788 | 0.1510 | 0.1423 | 0.1031 | 0.1352 | 0.1330 | 0.1236 |
| Jilin | 0.0664 | 0.0653 | 0.0634 | 0.0638 | 0.0621 | 0.0457 | 0.0563 | 0.0582 | 0.0495 |
| Heilongjiang | 0.0952 | 0.0957 | 0.0787 | 0.0720 | 0.0713 | 0.0434 | 0.0612 | 0.0626 | 0.0561 |
| Shanghai | 0.3214 | 0.3964 | 0.3695 | 0.3773 | 0.3444 | 0.2458 | 0.3451 | 0.3607 | 0.3851 |
| Jiansu | 0.5925 | 0.5997 | 0.5759 | 0.5620 | 0.5367 | 0.4088 | 0.5384 | 0.5341 | 0.5142 |
| Zhejiang | 0.4422 | 0.4456 | 0.4694 | 0.4761 | 0.4559 | 0.3495 | 0.4815 | 0.4817 | 0.4680 |
| Anhui | 0.1341 | 0.1520 | 0.1601 | 0.1613 | 0.1610 | 0.1229 | 0.1790 | 0.1873 | 0.2020 |
| Fujian | 0.1859 | 0.1938 | 0.2113 | 0.2410 | 0.2696 | 0.1877 | 0.2341 | 0.2036 | 0.2307 |
| Jiangxi | 0.0791 | 0.0856 | 0.0941 | 0.0879 | 0.0993 | 0.3001 | 0.1174 | 0.1267 | 0.1219 |
| Shandong | 0.4675 | 0.4412 | 0.3981 | 0.4190 | 0.4117 | 0.3161 | 0.3493 | 0.3688 | 0.4021 |
| Henan | 0.1606 | 0.1786 | 0.1878 | 0.1949 | 0.1901 | 0.1484 | 0.2009 | 0.2051 | 0.2026 |
| Hubei | 0.1701 | 0.1859 | 0.2021 | 0.2011 | 0.1941 | 0.1397 | 0.2056 | 0.2060 | 0.2035 |
| Hunan | 0.1282 | 0.1415 | 0.1420 | 0.1502 | 0.1501 | 0.1188 | 0.1617 | 0.1715 | 0.1764 |
| Guangdong | 0.7814 | 0.7702 | 0.7151 | 0.7461 | 0.7299 | 0.5995 | 0.7943 | 0.7974 | 0.7919 |
| Guangxi | 0.0671 | 0.0771 | 0.0741 | 0.0788 | 0.0758 | 0.2744 | 0.0940 | 0.1022 | 0.1111 |
| Hainan | 0.0450 | 0.0543 | 0.0595 | 0.0542 | 0.0514 | 0.0320 | 0.0519 | 0.0483 | 0.0449 |
| Chongqing | 0.0928 | 0.1139 | 0.1141 | 0.1240 | 0.1203 | 0.0964 | 0.1292 | 0.1354 | 0.1345 |
| Sichuan | 0.1791 | 0.2047 | 0.2085 | 0.2081 | 0.2169 | 0.1755 | 0.2404 | 0.2509 | 0.2426 |
| Guizhou | 0.0514 | 0.0538 | 0.0601 | 0.0678 | 0.0696 | 0.0565 | 0.0806 | 0.0823 | 0.0984 |
| Yunnan | 0.0775 | 0.0815 | 0.0822 | 0.0802 | 0.0789 | 0.0605 | 0.0875 | 0.0929 | 0.0858 |
| Xizang | 0.0140 | 0.0193 | 0.0200 | 0.0254 | 0.0172 | 0.0096 | 0.0115 | 0.0102 | 0.0100 |
| Shannxi | 0.1235 | 0.1355 | 0.1364 | 0.1425 | 0.1385 | 0.1064 | 0.1541 | 0.1538 | 0.1521 |
| Gansu | 0.0426 | 0.0473 | 0.0505 | 0.0450 | 0.0424 | 0.0325 | 0.0417 | 0.0438 | 0.0437 |
| Qinghai | 0.0175 | 0.0161 | 0.0239 | 0.0239 | 0.0200 | 0.0176 | 0.0189 | 0.0193 | 0.0222 |
| Ningxia | 0.0281 | 0.0316 | 0.0332 | 0.0310 | 0.0293 | 0.0231 | 0.0263 | 0.0244 | 0.0249 |
| Xinjiang | 0.0460 | 0.0477 | 0.0489 | 0.0392 | 0.0355 | 0.0336 | 0.0435 | 0.0445 | 0.0465 |
| Mean value | 0.1825 | 0.1915 | 0.1895 | 0.1908 | 0.1859 | 0.1554 | 0.1895 | 0.1922 | 0.1944 |

## Measurement results of the SBM model

Table 6 illustrates that China's agricultural carbon emission efficiency has exhibited a general upward trend with fluctuations, indicating notable outcomes in China's agricultural carbon emission reduction endeavors. This is closely associated with China's increased investment in agricultural environmental management, effective protection of agricultural land, and robust promotion of the policy of returning farmland to forests. From the perspective of individual provinces, Beijing, Shanghai, and Fujian exhibit the highest agricultural input-output efficiency, with agricultural carbon emission efficiency reaching 1 over the long term. Inner Mongolia exhibits the second-highest efficiency, failing to reach 1 only in 2010 and 2015, but maintaining high efficiency overall. Chongqing has consistently demonstrated a high level of efficiency, while Yunnan, Gansu, Qinghai, and Ningxia have exhibited a stable level of moderate efficiency. Shaanxi and Guizhou have exhibited a similar pattern, with their agricultural carbon emission efficiency initially at a moderate level before 2010, but subsequently

**Table 6. Results of SBM measurements.**

| Province | 2013 | 2014 | 2015 | 2016 | 2017 | 2018 | 2019 | 2020 | 2021 |
|---|---|---|---|---|---|---|---|---|---|
| Beijing | 1.0000 | 1.0000 | 1.0000 | 1.0000 | 1.0000 | 1.0000 | 1.0000 | 1.0000 | 0.7587 |
| Tianjin | 0.4265 | 0.4088 | 0.4796 | 0.4603 | 0.4372 | 0.4432 | 0.4262 | 0.4836 | 0.4745 |
| Hebei | 0.3494 | 0.3170 | 0.3361 | 0.3326 | 0.3166 | 0.3213 | 0.3077 | 0.3342 | 0.3230 |
| Shanxi | 0.2266 | 0.2138 | 0.2304 | 0.2400 | 0.2328 | 0.2178 | 0.2151 | 0.2400 | 0.2427 |
| Neimenggu | 0.3050 | 0.2782 | 0.2924 | 0.2685 | 0.2438 | 0.2410 | 0.2276 | 0.2302 | 0.2236 |
| Liaoning | 0.6118 | 0.5545 | 0.6039 | 0.5091 | 0.4864 | 0.4621 | 0.4385 | 0.4213 | 0.4216 |
| Jilin | 0.2861 | 0.2570 | 0.2696 | 0.2205 | 0.1907 | 0.1838 | 0.1811 | 0.2031 | 0.1827 |
| Heilongjiang | 0.4259 | 0.3972 | 0.4445 | 0.4150 | 0.4060 | 0.3727 | 0.3573 | 0.3491 | 0.3114 |
| Shanghai | 1.0000 | 1.0000 | 1.0000 | 1.0000 | 0.6575 | 0.6505 | 0.5906 | 0.5495 | 0.5027 |
| Jiansu | 0.5976 | 0.5693 | 1.0000 | 1.0000 | 0.5859 | 0.5360 | 0.5044 | 0.4826 | 0.4701 |
| Zhejiang | 0.7181 | 0.6818 | 1.0000 | 1.0000 | 1.0000 | 1.0000 | 1.0000 | 1.0000 | 1.0000 |
| Anhui | 0.2460 | 0.2351 | 0.2612 | 0.2523 | 0.2454 | 0.2242 | 0.2209 | 0.2598 | 0.2541 |
| Fujian | 1.0000 | 1.0000 | 1.0000 | 1.0000 | 1.0000 | 1.0000 | 1.0000 | 1.0000 | 1.0000 |
| Jiangxi | 0.3830 | 0.3680 | 0.4164 | 0.4062 | 0.3789 | 0.3672 | 0.3736 | 0.3815 | 0.3675 |
| Shandong | 0.4229 | 0.4060 | 0.4593 | 0.4209 | 0.3918 | 0.3730 | 0.3471 | 0.3380 | 0.3587 |
| Henan | 0.2703 | 0.2602 | 0.2895 | 0.2754 | 0.2722 | 0.2567 | 0.2552 | 0.2758 | 0.2700 |
| Hubei | 0.3929 | 0.3741 | 0.4283 | 0.4226 | 0.4010 | 0.3741 | 0.3725 | 0.3945 | 0.4173 |
| Hunan | 0.3258 | 0.3002 | 0.3279 | 0.3231 | 0.3048 | 0.2853 | 0.3187 | 0.3951 | 0.3681 |
| Guangdong | 0.6305 | 0.6016 | 0.6902 | 0.6811 | 0.6466 | 0.6543 | 0.6901 | 0.7054 | 0.6897 |
| Guangxi | 0.3490 | 0.3314 | 0.3721 | 0.3628 | 0.3398 | 0.3233 | 0.3247 | 0.3559 | 0.3641 |
| Hainan | 0.8522 | 1.0000 | 1.0000 | 1.0000 | 1.0000 | 0.8763 | 1.0000 | 1.0000 | 1.0000 |
| Chongqing | 0.3247 | 0.3119 | 0.3656 | 0.3772 | 0.3581 | 0.3523 | 0.3632 | 0.3989 | 0.3906 |
| Sichuan | 0.4271 | 0.4026 | 0.4676 | 0.4514 | 0.4253 | 0.3995 | 0.4004 | 0.4443 | 0.4128 |
| Guizhou | 0.2450 | 0.2824 | 0.4046 | 0.4584 | 0.5296 | 0.5109 | 0.5988 | 1.0000 | 1.0000 |
| Yunnan | 0.2818 | 0.2687 | 0.2955 | 0.2747 | 0.2646 | 0.2849 | 0.3184 | 0.3597 | 0.3624 |
| Xizang | 0.2548 | 0.2581 | 0.2790 | 0.3195 | 0.3152 | 0.3203 | 0.3260 | 0.3225 | 0.3246 |
| Shannxi | 0.3747 | 0.3654 | 0.4015 | 0.3807 | 0.3508 | 0.3316 | 0.3327 | 0.3843 | 0.3790 |
| Gansu | 0.1950 | 0.1837 | 0.2058 | 0.2166 | 0.2247 | 0.2164 | 0.2199 | 0.2283 | 0.2478 |
| Qinghai | 0.3594 | 0.3441 | 0.3453 | 1.0000 | 1.0000 | 1.0000 | 1.0000 | 1.0000 | 1.0000 |
| Ningxia | 0.2343 | 0.2287 | 0.2733 | 0.2696 | 0.2616 | 0.2750 | 0.2467 | 0.2767 | 0.2736 |
| Xinjiang | 0.3917 | 0.3628 | 0.3914 | 0.3437 | 0.3304 | 0.3238 | 0.3058 | 0.3178 | 0.3517 |
| Mean value | 0.4486 | 0.4375 | 0.4945 | 0.5059 | 0.4709 | 0.4573 | 0.4601 | 0.4881 | 0.4756 |

improving significantly, indicating greater progress in emission reduction. In contrast, Tibet's agricultural carbon emission efficiency was at the production frontier between 2006 and 2012, but began to decline precipitously to the medium level in 2013.

## Degree of coupling and coordination

The level of coupling and coordination between the digital economy and agricultural carbon emissions performance in China was measured using the index presented in Table 7. As shown in the table, the coupling degree exhibited a decreasing trend from 0.8085 to 0 during the period of 2013–2021, followed by an increasing trend. From 2011 to 2018, the coupling coordination index remained stable at 0.7793, but increased slightly to 0.8099 from 2018 to 2021. When ranked by subregion, the coupling coordination index is highest in the Central region (0.9175), followed by the East region (0.8757), the national average (0.8099), the Northeast region (0.7921), and the West region (0.7058). This indicates that the East and Central

**Table 7. Degree of coupling.**

| Province | 2013 | 2014 | 2015 | 2016 | 2017 | 2018 | 2019 | 2020 | 2021 |
|---|---|---|---|---|---|---|---|---|---|
| Beijing | 0.9720 | 0.9786 | 0.9824 | 0.9789 | 0.9759 | 0.9349 | 0.9755 | 0.9764 | 0.9981 |
| Tianjin | 0.8757 | 0.8964 | 0.8549 | 0.8417 | 0.8177 | 0.7193 | 0.8156 | 0.7944 | 0.7920 |
| Hebei | 0.9103 | 0.9334 | 0.9134 | 0.9246 | 0.9391 | 0.8762 | 0.9375 | 0.9287 | 0.9362 |
| Shanxi | 0.8802 | 0.8942 | 0.8512 | 0.8500 | 0.8493 | 0.8178 | 0.8733 | 0.8526 | 0.8549 |
| Neimenggu | 0.7462 | 0.7601 | 0.7758 | 0.8017 | 0.8089 | 0.7121 | 0.8053 | 0.8147 | 0.8347 |
| Liaoning | 0.8557 | 0.8776 | 0.8396 | 0.8401 | 0.8369 | 0.7723 | 0.8488 | 0.8542 | 0.8373 |
| Jilin | 0.7822 | 0.8038 | 0.7850 | 0.8344 | 0.8608 | 0.7985 | 0.8506 | 0.8323 | 0.8193 |
| Heilongjiang | 0.7727 | 0.7911 | 0.7150 | 0.7100 | 0.7130 | 0.6113 | 0.7065 | 0.7180 | 0.7194 |
| Shanghai | 0.8581 | 0.9018 | 0.8877 | 0.8919 | 0.9499 | 0.8923 | 0.9650 | 0.9782 | 0.9912 |
| Jiansu | 1.0000 | 0.9997 | 0.9631 | 0.9599 | 0.9990 | 0.9909 | 0.9995 | 0.9987 | 0.9990 |
| Zhejiang | 0.9713 | 0.9778 | 0.9325 | 0.9349 | 0.9275 | 0.8762 | 0.9368 | 0.9368 | 0.9320 |
| Anhui | 0.9557 | 0.9767 | 0.9708 | 0.9755 | 0.9782 | 0.9564 | 0.9945 | 0.9868 | 0.9934 |
| Fujian | 0.7272 | 0.7375 | 0.7590 | 0.7911 | 0.8180 | 0.7295 | 0.7841 | 0.7498 | 0.7805 |
| Jiangxi | 0.7534 | 0.7825 | 0.7755 | 0.7650 | 0.8112 | 0.9949 | 0.8530 | 0.8653 | 0.8649 |
| Shandong | 0.9987 | 0.9991 | 0.9974 | 1.0000 | 0.9997 | 0.9966 | 1.0000 | 0.9990 | 0.9984 |
| Henan | 0.9671 | 0.9825 | 0.9770 | 0.9853 | 0.9841 | 0.9636 | 0.9929 | 0.9891 | 0.9898 |
| Hubei | 0.9184 | 0.9418 | 0.9334 | 0.9348 | 0.9376 | 0.8898 | 0.9574 | 0.9494 | 0.9389 |
| Hunan | 0.9004 | 0.9332 | 0.9184 | 0.9308 | 0.9404 | 0.9111 | 0.9451 | 0.9189 | 0.9360 |
| Guangdong | 0.9943 | 0.9924 | 0.9998 | 0.9990 | 0.9982 | 0.9990 | 0.9975 | 0.9981 | 0.9976 |
| Guangxi | 0.7357 | 0.7825 | 0.7441 | 0.7658 | 0.7722 | 0.9966 | 0.8345 | 0.8327 | 0.8464 |
| Hainan | 0.4367 | 0.4419 | 0.4605 | 0.4418 | 0.4313 | 0.3685 | 0.4332 | 0.4192 | 0.4054 |
| Chongqing | 0.8316 | 0.8853 | 0.8516 | 0.8631 | 0.8677 | 0.8214 | 0.8798 | 0.8700 | 0.8730 |
| Sichuan | 0.9124 | 0.9454 | 0.9237 | 0.9295 | 0.9459 | 0.9210 | 0.9683 | 0.9606 | 0.9657 |
| Guizhou | 0.7574 | 0.7334 | 0.6713 | 0.6701 | 0.6407 | 0.5987 | 0.6467 | 0.5301 | 0.5711 |
| Yunnan | 0.8226 | 0.8452 | 0.8252 | 0.8365 | 0.8414 | 0.7603 | 0.8225 | 0.8078 | 0.7870 |
| Xizang | 0.4442 | 0.5086 | 0.5000 | 0.5225 | 0.4429 | 0.3366 | 0.3633 | 0.3446 | 0.3399 |
| Shannxi | 0.8635 | 0.8884 | 0.8701 | 0.8904 | 0.9009 | 0.8576 | 0.9302 | 0.9036 | 0.9042 |
| Gansu | 0.7672 | 0.8069 | 0.7954 | 0.7550 | 0.7312 | 0.6739 | 0.7323 | 0.7352 | 0.7141 |
| Qinghai | 0.4207 | 0.4130 | 0.4917 | 0.3021 | 0.2775 | 0.2610 | 0.2701 | 0.2729 | 0.2914 |
| Ningxia | 0.6183 | 0.6531 | 0.6215 | 0.6081 | 0.6020 | 0.5346 | 0.5897 | 0.5456 | 0.5532 |
| Xinjiang | 0.6132 | 0.6410 | 0.6284 | 0.6061 | 0.5921 | 0.5837 | 0.6603 | 0.6566 | 0.6422 |

Note: On June 13, 2011, the National Bureau of Statistics (NBS) categorized China's economic regions into four regions: Eastern, Central, Western, and Northeastern. The Eastern region includes Beijing, Tianjin, Hebei, Shanghai, Jiangsu, Zhejiang, Fujian, Shandong, Guangdong, and Hainan. The Central region includes Shanxi, Anhui, Jiangxi, Henan, Hubei, and Hunan. The Western region includes Inner Mongolia, Guangxi, Chongqing, Sichuan, Guizhou, Yunnan, Tibet, Shaanxi, Gansu, Qinghai, Ningxia, and Xinjiang. The Northeastern region includes Liaoning, Jilin, and Heilongjiang.

regions have higher coupling coordination than the national average, while the West and Northeast regions have lower coupling coordination than the national average.

Table 8 displays the degree of coupling and coordination between China's digital economy and agricultural carbon emission performance. It is evident that the coupling coordination degree between the two exhibits a decreasing and then increasing trend, from 0.4885 to 0.4689 during the period of 2013–2018. Specifically, the coupling coordination degree of the eastern, central, western, and northeastern regions is as follows: eastern (0.6730) > national average (0.4955) > central (0.4547) > northeastern (0.4209) > west(0.3866). The Eastern region has a higher coupling coordination degree than the national average, while the Central, Western,

**Table 8. Coupling and coordination degree.**

| Province | 2013 | 2014 | 2015 | 2016 | 2017 | 2018 | 2019 | 2020 | 2021 |
|---|---|---|---|---|---|---|---|---|---|
| Beijing | 0.8872 | 0.9008 | 0.9098 | 0.9016 | 0.8951 | 0.8306 | 0.8942 | 0.8962 | 0.8442 |
| Tianjin | 0.5019 | 0.5039 | 0.5196 | 0.5016 | 0.4763 | 0.4337 | 0.4693 | 0.4889 | 0.4831 |
| Hebei | 0.4743 | 0.4667 | 0.4671 | 0.4719 | 0.4704 | 0.4358 | 0.4626 | 0.4758 | 0.4731 |
| Shanxi | 0.3678 | 0.3634 | 0.3586 | 0.3655 | 0.3597 | 0.3362 | 0.3554 | 0.3666 | 0.3696 |
| Neimenggu | 0.3697 | 0.3580 | 0.3729 | 0.3671 | 0.3524 | 0.3175 | 0.3392 | 0.3445 | 0.3469 |
| Liaoning | 0.5874 | 0.5735 | 0.5732 | 0.5266 | 0.5129 | 0.4672 | 0.4934 | 0.4866 | 0.4777 |
| Jilin | 0.3713 | 0.3599 | 0.3615 | 0.3444 | 0.3298 | 0.3027 | 0.3177 | 0.3298 | 0.3084 |
| Heilongjiang | 0.4487 | 0.4415 | 0.4325 | 0.4158 | 0.4125 | 0.3566 | 0.3845 | 0.3844 | 0.3636 |
| Shanghai | 0.7530 | 0.7935 | 0.7797 | 0.7837 | 0.6898 | 0.6324 | 0.6719 | 0.6672 | 0.6633 |
| Jiansu | 0.7714 | 0.7644 | 0.8711 | 0.8658 | 0.7488 | 0.6842 | 0.7219 | 0.7125 | 0.7012 |
| Zhejiang | 0.7507 | 0.7424 | 0.8277 | 0.8307 | 0.8217 | 0.7689 | 0.8330 | 0.8331 | 0.8271 |
| Anhui | 0.4262 | 0.4348 | 0.4522 | 0.4491 | 0.4459 | 0.4074 | 0.4459 | 0.4697 | 0.4760 |
| Fujian | 0.6567 | 0.6635 | 0.6780 | 0.7006 | 0.7206 | 0.6582 | 0.6956 | 0.6718 | 0.6930 |
| Jiangxi | 0.4172 | 0.4213 | 0.4449 | 0.4347 | 0.4404 | 0.5761 | 0.4576 | 0.4689 | 0.4600 |
| Shandong | 0.6668 | 0.6506 | 0.6539 | 0.6480 | 0.6337 | 0.5860 | 0.5901 | 0.5942 | 0.6163 |
| Henan | 0.4565 | 0.4643 | 0.4829 | 0.4813 | 0.4769 | 0.4418 | 0.4759 | 0.4877 | 0.4836 |
| Hubei | 0.5085 | 0.5135 | 0.5424 | 0.5399 | 0.5282 | 0.4781 | 0.5261 | 0.5339 | 0.5398 |
| Hunan | 0.4521 | 0.4540 | 0.4645 | 0.4693 | 0.4625 | 0.4290 | 0.4764 | 0.5102 | 0.5048 |
| Guangdong | 0.8378 | 0.8250 | 0.8382 | 0.8443 | 0.8289 | 0.7914 | 0.8604 | 0.8660 | 0.8597 |
| Guangxi | 0.3913 | 0.3998 | 0.4074 | 0.4112 | 0.4006 | 0.5458 | 0.4180 | 0.4368 | 0.4485 |
| Hainan | 0.4426 | 0.4826 | 0.4939 | 0.4826 | 0.4762 | 0.4091 | 0.4773 | 0.4688 | 0.4602 |
| Chongqing | 0.4167 | 0.4342 | 0.4520 | 0.4651 | 0.4556 | 0.4293 | 0.4654 | 0.4821 | 0.4788 |
| Sichuan | 0.5259 | 0.5358 | 0.5588 | 0.5536 | 0.5511 | 0.5146 | 0.5570 | 0.5779 | 0.5625 |
| Guizhou | 0.3350 | 0.3511 | 0.3950 | 0.4199 | 0.4381 | 0.4121 | 0.4687 | 0.5356 | 0.5600 |
| Yunnan | 0.3844 | 0.3847 | 0.3948 | 0.3853 | 0.3802 | 0.3624 | 0.4086 | 0.4276 | 0.4200 |
| Xizang | 0.2443 | 0.2656 | 0.2734 | 0.3002 | 0.2713 | 0.2357 | 0.2476 | 0.2394 | 0.2384 |
| Shannxi | 0.4638 | 0.4717 | 0.4837 | 0.4826 | 0.4695 | 0.4334 | 0.4758 | 0.4931 | 0.4900 |
| Gansu | 0.3019 | 0.3053 | 0.3192 | 0.3143 | 0.3125 | 0.2896 | 0.3095 | 0.3163 | 0.3226 |
| Qinghai | 0.2816 | 0.2727 | 0.3013 | 0.3933 | 0.3762 | 0.3644 | 0.3710 | 0.3729 | 0.3859 |
| Ningxia | 0.2848 | 0.2916 | 0.3086 | 0.3023 | 0.2959 | 0.2823 | 0.2837 | 0.2866 | 0.2873 |
| Xinjiang | 0.3663 | 0.3627 | 0.3719 | 0.3406 | 0.3291 | 0.3230 | 0.3396 | 0.3449 | 0.3576 |
| Mean value | 0.4885 | 0.4920 | 0.5094 | 0.5095 | 0.4956 | 0.4689 | 0.4933 | 0.5023 | 0.5001 |

and Northeastern regions have a lower coupling coordination degree than the national average. The minimum coupling coordination degree grade for each province is 0.2357, and the maximum is 0.9098. The full potential of synergies between Chinese provinces has not been realized yet. There is still significant room for improvement in the degree of coupling and coordination between the digital economy and agricultural carbon emission performance.

(1) Spatial distribution of coupling degree

In this paper, we categorize the coupling C values into four intervals: [0, 0.3] for low coupling, [0.3, 0.5] for antagonistic phase, [0.5, 0.8] for friction phase, and [0.8, 1] for high coupling. To illustrate the distribution of coupling more clearly, we utilized ArcGIS10.5 to display the spatial layout of coupling for representative years, as shown in Fig 2. As shown in Fig 2, from 2013 to 2021, all provinces are in the grinding stage and high level stage, indicating that the coupling degree between digital economy and agricultural carbon emission performance is high in China. This is mainly due to:

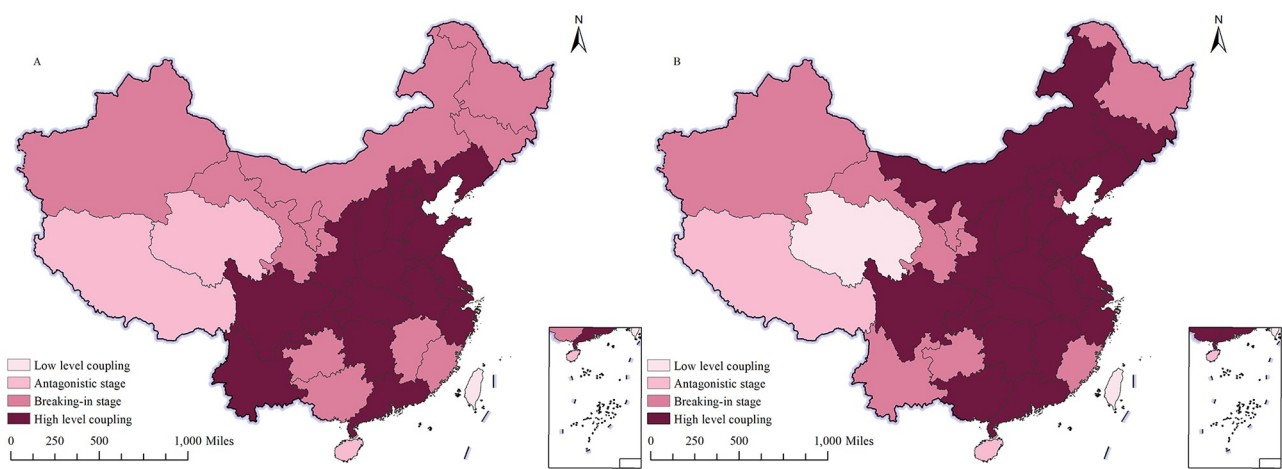

**Fig 2. Spatial distribution of coupling degree.** A shows the spatial distribution of coupling degree in 2013; B shows the spatial distribution of coupling degree in 2021. The map is obtained from Natural Earth (http://www.naturalearthdata.com/).

Firstly, the development of the digital economy can facilitate the restructuring of the agricultural carbon emission industry and enhance efficiency, thereby enhancing the performance of agricultural carbon emissions. Secondly, the penetration depth of digital technology in the field of agricultural production requires strengthening, and its integration with the agricultural sector still lacks universality. In 2013, there were three provinces in the antagonistic stage, ten in the grinding stage, and eighteen in the high-level coupling. In 2021, there was one province in the low coupling stage, two provinces in the antagonistic stage, eight in the grinding stage, and twenty in the high-level coupling stage. The number of provinces in the high coupling stage has increased, while the number of provinces in the antagonistic and grinding stages has decreased. Overall, there is a coupling between the digital economy and the performance of agricultural carbon emissions, and the level of coupling has increased. With regard to spatial differentiation, the regional distribution of low and medium coupling exhibits clear spatial clustering characteristics and maintains stability in inter-annual changes. The coupling degree of the northeastern region of China is gradually decreasing, primarily due to constraints posed by the level of local economic development and limited breadth and depth of digital technology application.

(2) Spatial distribution of the degree of coupling coordination This paper categorizes the coupling and coordination degree of the digital economy and agricultural carbon emission performance into four levels, as indicated in Fig 3. The levels are defined as follows: low (0–0.3), moderate (0.3–0.5), high (0.5–0.8), and extreme (0.8–1). The figure shows that the coupling and coordination degree of the digital economy and agricultural carbon emission performance exhibit regional heterogeneity. The spatial distribution characteristics indicate that the eastern coastal provinces have achieved highly coordinated development, while the central, western, and northeastern regions are dominated by a medium degree of coordination and coupling. The various regions are converging towards a phenomenon of balanced development, with high levels in the east and south and low levels in the west and north. In terms of temporal trends, the central region has gradually transitioned from low coupling coordination to medium coupling coordination. Significant progress has been made, particularly in the southwest of China. However, the spatial distribution characteristics of "high in the east and low in the west, high in the south and low in the north" remain unchanged. Regions with high

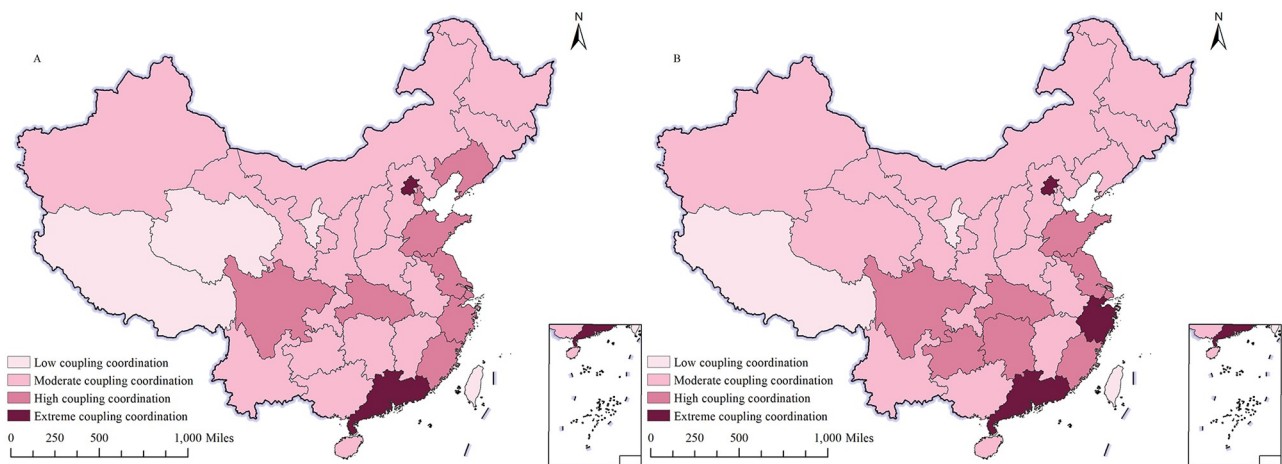

**Fig 3. Spatial distribution of the degree of coupling coordination.** A shows the Spatial distribution of the degree of coupling coordination in 2013; B shows the Spatial distribution of the degree of coupling coordination in 2021. The map is obtained from Natural Earth (http://www.naturalearthdata.com/).

and medium coordination continue to concentrate and show a trend of diffusion, with a belt-shaped distribution. The distribution, previously described as "point-like," has evolved to become "slice-like" or "belt-like".

The spatial divergence of the performance of the digital economy and agricultural carbon emissions is influenced by a number of factors:

Firstly, the eastern coastal regions are more advanced in the application of digital technology, which has resulted in a higher degree of digital technology application and a more developed digital economy. However, digital technology applications in some inland or rural areas are limited due to infrastructure and technological constraints. As a result, the development of the digital economy in these areas lags behind.

Secondly, certain regions have a unique agricultural industrial structure, with a primary focus on traditional agriculture and a lack of modern agricultural technology and resources, resulting in lower agricultural carbon emission performance. However, in some regions, the agricultural industrial structure is more diversified, with a focus on modern agriculture and the application of digital technology. This has led to a faster development of the digital economy and relatively high agricultural carbon emission performance.

Thirdly, in certain regions, the government provides more robust support for the digital economy, offering preferential policies and financial assistance. These policies facilitate the adoption of digital technologies in agriculture and enhance carbon emission performance. However, certain regions receive less policy support and have limited implementation of digital technology in agriculture, resulting in lower agricultural carbon emission performance.

## Spatial econometric analysis

Table 9 presents the global Moran index of the coupled coordination degree of the digital economy and agricultural carbon emission performance, as analyzed using ArcGIS and Stata. The results indicate that the coupling and coordination degree of the digital economy and agricultural carbon emission performance have passed the significance test at the 1% level, with all values being positive. Overall, the degree of coupling and coordination between the digital economy and agricultural carbon emission performance is primarily influenced by

**Table 9. Global Moran index.**

| Year | Index | p-value* |
|------|-------|----------|
| 2013 | 0.345*** | 0.001 |
| 2014 | 0.373*** | 0.001 |
| 2015 | 0.397*** | 0.000 |
| 2016 | 0.399*** | 0.000 |
| 2017 | 0.367*** | 0.001 |
| 2018 | 0.408*** | 0.000 |
| 2019 | 0.358*** | 0.001 |
| 2020 | 0.349*** | 0.001 |
| 2021 | 0.369*** | 0.001 |

Note: The symbols ***, **, and * represent statistical significance at the 1%, 5%, and 10% levels, respectively.

neighboring provinces. This results in a geographic clustering pattern where high values are adjacent to high values and low values are adjacent to low values, indicating a clear positive correlation. The main reason for this is:

(1) The technological levels of different regions vary, which affects the development of the digital economy and agricultural carbon emissions performance. Regions with strong R&D capabilities and technology accumulation in digital and agricultural technology have an advantage in the development of the digital economy and agricultural carbon emissions performance. However, some regions lack the necessary technical talent and research and development capabilities, which limits the application of digital and agricultural technology. Consequently, this impedes the advancement of the digital economy and the enhancement of agricultural carbon emission performance.

(2) Regional variations in industrial structure also influence the advancement of the digital economy and the carbon footprint of agriculture. In areas with a more diverse industrial base, there is a heightened focus on the advancement of modern agriculture and the integration of digital technology, which has led to a rapid expansion of the digital economy and a higher agricultural carbon footprint. In certain regions, the industrial structure is relatively uniform, with a primary reliance on traditional agriculture. The lack of modern agricultural technology and resources results in a low agricultural carbon emission performance.

(3) The "Point-Axis Development Model" is a theoretical framework that elucidates the radiation effect of economic development. It was proposed by Polish economists Zaremba and Malisz. The core idea is to form a core area in the region with favorable location conditions and present a point distribution. In the subsequent period, the core areas continue to expand and connect with each other to form an axis, which radiates the surrounding area and forms a new core area. When the overall development level of a certain region is enhanced, the technology spillover effect can rapidly disseminate to the surrounding areas, thereby forming a cluster in a specific area.

The spatial distribution of the local Moran index in 2013 and 2021 is presented in Fig 4. As illustrated in Fig 4, there has been no discernible change in the range of high-high and low-high spatial clustering. The spatial distribution of the coordinated coordination degree of the digital economy and agricultural carbon emission performance is more balanced, exhibiting

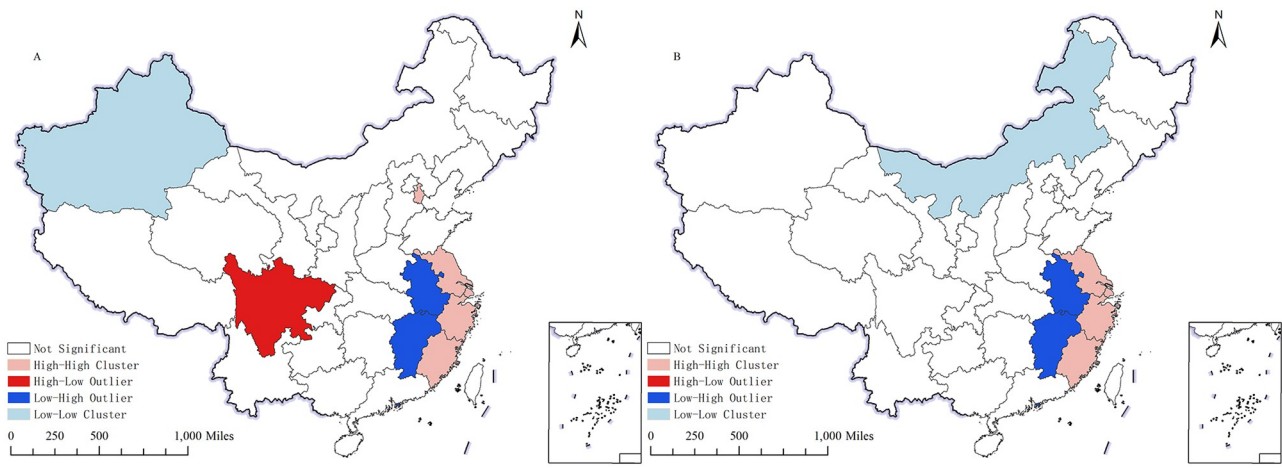

**Fig 4. LISA clustering diagram.** A shows the LISA clustering diagram of degree of coupling coordination in 2013; B shows the LISA clustering diagram of degree of coupling coordination in 2021. The map is obtained from Natural Earth (http://www.naturalearthdata.com/).

positive spatial auto-correlation (high-high and low-low) and negative spatial auto-correlation (high-low and low-high).

## Discussion

The objective of this study is to examine the extent of coupling, coordination, and spatio-temporal heterogeneity between the digital economy and agricultural carbon emission performance. The results of this study indicate a significant positive relationship between the digital economy and agricultural carbon emissions performance. This implies that the advancement of the digital economy can potentially reduce agricultural carbon emissions. Additionally, this study has identified significant spatial and temporal heterogeneity in the degree of coupling and coordination between the digital economy and agricultural carbon emission performance across different regions.

The study's conclusions are significant for understanding and responding to climate change, as well as promoting sustainable development. The digital economy has become a crucial force in promoting economic growth and improving productivity, driven by the continuous development and popularization of digital technology. Simultaneously, the development of the digital economy can promote the transformation of the energy structure and improve energy utilization efficiency, thereby reducing carbon emissions. Therefore, the digital economy can be an important means of promoting the development of a low-carbon economy.

### Relationship between the digital economy and agricultural carbon performance

The results of the study indicate a significant positive correlation between the digital economy and the reduction of agricultural carbon emissions. This finding is consistent with previous research that suggests that digital technology can enhance agricultural productivity, reduce energy consumption, and lower carbon emissions in agricultural production.

Digital technologies can enhance agricultural carbon performance in a number of ways. For instance, they can enhance agricultural production efficiency, reduce energy consumption, and lower carbon emissions per unit of output. Additionally, digital technology can facilitate the digital transformation of the agricultural industry chain, improve the energy utilization

efficiency of the agricultural industry chain, and decrease carbon emissions. Furthermore, digital technology can promote the use of renewable energy and reduce energy consumption and carbon emissions in agricultural production.

Nevertheless, the advent of the digital economy may also result in an uptick in agricultural carbon emissions. The proliferation of digital technologies may prompt shifts in agricultural land use, which could influence carbon emissions performance. For instance, the advent of e-commerce could lead to elevated energy consumption and carbon emissions in sectors such as logistics and distribution. Similarly, the advent of digital services like online education could prompt an increase in the number of trips taken, thereby boosting carbon emissions in the transportation sector.

There is a notable degree of spatial and temporal heterogeneity in the degree of coupling and coordination between the digital economy and agricultural carbon emission performance in different regions. In certain regions, the advancement of the digital economy has led to a reduction in agricultural carbon emissions through the effective application of digital technology. However, in other regions where the digital economy is less developed, the impact of digital technology on agricultural carbon emissions may be limited. Consequently, in order to facilitate the advancement of the digital economy, it is essential to devise policies and measures that are specifically tailored to the distinctive circumstances of different regions. This will facilitate the integration and collaboration between the digital economy and agricultural carbon emission performance.

## Analysis of spatio-temporal heterogeneity

The degree of coupling and coordination between the digital economy and agricultural carbon emission performance varies across regions, indicating significant spatial and temporal heterogeneity. This finding suggests that the relationship between the digital economy and agricultural carbon emission performance is influenced by a variety of factors, including policy, technology, and the economy. There are significant differences in policy environment, technology level, economic development, and other factors among regions that affect the degree of coupling and coordination between the digital economy and agricultural carbon emissions performance.

In eastern regions, the digital economy has been extensively utilized in the agricultural sector, optimizing the agricultural production process and improving energy efficiency throughout the agricultural industry chain. In contrast, the digital economy is less developed in western China, and the application of digital technology in agriculture is limited in these regions. Consequently, the impact of the digital economy on the performance of agricultural carbon emissions is relatively small.

In terms of temporality, there are considerable discrepancies in the extent of coupling and coordination between the digital economy and agricultural carbon emission performance. As digital technology continues to evolve and become more pervasive, the relationship between the digital economy and agricultural carbon emission performance is also undergoing transformation. For instance, during the nascent stages of integrating digital technology into agriculture, the impact of the digital economy on agricultural carbon emission performance may be constrained by the technology's immaturity and high cost.

The degree of coupling and coordination between the digital economy and agricultural carbon emission performance exhibits significant spatio-temporal heterogeneity. To fully understand this phenomenon, it is essential to consider the influence of multiple factors and adopt appropriate data and methods. Future research should aim to further investigate the spatial and temporal heterogeneity between the digital economy and agricultural carbon emissions

performance. This will provide a scientific basis for promoting the development of low-carbon agriculture.

## Limitations of this study

This study is subject to certain limitations that would benefit from further expansion and depth in future research.

(1) This study is primarily dependent on official statistics, which may influence the results to a significant extent. However, errors and biases in data collection and processing may impact the accuracy and reliability of the findings.

(2) This study solely focuses on the degree of coupling coordination between the digital economy and agricultural carbon emission performance, without taking into account other factors. For example, the development of the digital economy may have adverse effects on the agro-ecological environment, such as the digital divide, data security, and other related concerns. Therefore, future studies should comprehensively evaluate the impact of the digital economy on the economy, society, and the environment to more accurately assess its effect on sustainable development.

(3) The models and methods used in this study may have limitations. For instance, the index system employed to assess the coupling coordination degree may exhibit deficiencies that fail to fully capture the interrelationship and impact of the digital economy on agricultural carbon emission performance. Furthermore, it is essential to acknowledge that the regression analysis and other methodologies utilized in this study may possess inherent limitations and may not fully account for the influence of other factors.

## Conclusions and recommendations

This paper employs provincial panel data from 2013 to 2021 to assess the level of digital economy development in each province. It employs the SBM to measure the efficiency of agricultural carbon emissions. Based on these measurements, the coupling, coordination, and spatial-temporal divergence characteristics of the two are evaluated. The study results indicate that the digital economy of each province exhibited varying degrees of growth from 2013 to 2021. Nevertheless, there was a discernible tendency for differentiation in the development of the digital economy between provinces. Additionally, the average value of China's agricultural carbon emission efficiency has demonstrated a broad upward trend with fluctuations. Furthermore, from 2013 to 2021, China's digital economy and agricultural carbon emission performance exhibited a high degree of coupling. The coupling and coordination degrees of the digital economy and agricultural carbon emission performance exhibited a decreasing trend followed by an increasing trend.

The following recommendations are provided for consideration:

(1) Increased investment in digital economy infrastructure and technology
Promotion of balanced development of new agricultural digital infrastructure to achieve "double carbon". Upgrading of economic infrastructure and technology of agriculture, improvement of digitization, intelligence, and refinement of agricultural production, and promotion of deep integration of digital technology and agricultural production.

(2) The digitization of agriculture
It is recommended that agricultural digitization, including precision agriculture, smart

irrigation, drone plant protection, and agricultural big data analysis, be promoted in order to improve agricultural production efficiency and environmental protection. It is also important to pay attention to the integrated innovation and iterative upgrading of agricultural digitalization. It is imperative to leverage the advantages of the recently established national system to capitalize on the opportunities presented by the "14th Five-Year Plan." This entails prioritizing the advancement of agricultural science and technology innovation in pivotal areas such as energy conservation, emission reduction, and carbon reduction in agriculture and rural areas. This encompasses the reduction of fertilizer and pesticide usage, the utilization of livestock and poultry manure and crop residue resources, the development of renewable energy resources in rural areas, and the enhancement of digital agricultural machinery, processing, and circulation. To enhance scientific and technological innovation leadership, we will achieve breakthroughs in technological innovation through the unveiling system and establish a robust green agricultural digital system. We will accelerate the development of an intelligent Internet agricultural ecosystem, centered on an agricultural big data platform, and promote the growth of the Internet of Things in agriculture and a green low-carbon integrated technology service platform, among other initiatives. The integration and application of new information technologies, including 5G, big data, artificial intelligence, blockchain, and cloud computing, will be accelerated in order to promote the development of low-carbon agriculture.

(3) Strengthening policy guidance and financial support

The objective is to facilitate the adoption of digital technologies with the aim of enhancing agricultural productivity while promoting the utilisation of green technologies and clean energy for the development of low-carbon agriculture. It is recommended that leading agricultural enterprises leverage their endowment advantages and strive to become pioneers in agricultural emission reduction or zero emission. Agricultural emission reduction and zero emission incentive policies can be implemented, such as the provision of financial subsidies and tax incentives, to encourage agricultural head enterprises to actively participate in agricultural emission reduction and zero emission initiatives. Furthermore, it is possible to support the technological research and development and application of emission reduction and zero emission by agricultural head enterprises through government procurement and special funds. It is also necessary to strengthen the supervision and guidance of agricultural head enterprises and to promote the establishment of a sound emission reduction and zero-emission management system, with the aim of ensuring that they meet the emission reduction and zero-emission standards in their production processes.

(4) Establishing Coupled Coordination Mechanisms

The establishment of a coordination mechanism for the coupling of the digital economy and agricultural carbon emission performance is essential for the realization of the synergistic development of digital technology and agricultural production, as well as the promotion of the green transformation of agriculture. In order to ensure the smooth implementation of the agricultural "dual-carbon" strategy, it is necessary to improve the resilience of agricultural digital development. The focus of green and low-carbon development in agriculture and rural areas is on the development of independent core technology for agricultural digitization, with the objective of reducing carbon emissions and increasing profits. This will reinforce the scientific and technological foundation for digital agriculture and low-carbon development. In order to enhance the resilience of the synergistic development of agricultural digitization and the "dual-carbon" strategy, it is necessary to fully activate the ecological potential of data resources and digital technology. The construction of digital agriculture pilot zones covering the entire agricultural industry chain should be accelerated,

and a "green and low-carbon" benchmark agricultural science and technology demonstration park working system should be explored. It is necessary to examine the development of a benchmark agricultural science and technology demonstration park system that is environmentally friendly and has low carbon emissions.

(5) Enhancement of digital literacy and environmental awareness among farmers

The government can enhance farmers' digital literacy by offering digital training courses, lectures, and educational videos. Additionally, it can provide digital teaching materials and establish a digital service platform to offer digital services to farmers. For instance, a digital platform for agricultural information could be created to offer farmers services like agricultural information queries and technology consultations. Similarly, a digital platform for rural financial services could be established to provide financial assistance. These initiatives can enhance farmers' comprehension of digital knowledge, improve their digital literacy, and educate them on environmental awareness through practical activities. Organizing farmers to participate in environmental protection activities, such as garbage classification and tree planting, can help them develop good environmental habits. Using media resources, such as TV, radio, and newspapers, to promote the importance of environmental protection knowledge and digital literacy can increase farmers' awareness and motivation. By demonstrating the benefits of digital literacy and environmental awareness, advanced models can serve as a source of inspiration for other farmers, encouraging them to take an active role in improving their own digital literacy and environmental awareness.

## Supporting information

**S1 Data.**
(XLSX)

## Author Contributions

**Conceptualization:** Haisong Wang.

**Data curation:** Ning Zhu.

**Formal analysis:** Yuhuan Wu.

**Investigation:** Ning Zhu.

**Methodology:** Yuhuan Wu.

**Writing – original draft:** Haisong Wang.

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
