## [Decision Letter · Decision Letter 0]

17 Apr 2024

PONE-D-24-07374Digital economy and agricultural carbon emission performance coupling coordination measurement and spatio-temporal heterogeneity analysisPLOS ONE

Dear Dr. Wu,

Thank you for submitting your manuscript to PLOS ONE. After careful consideration, we feel that it has merit but does not fully meet PLOS ONE’s publication criteria as it currently stands. Therefore, we invite you to submit a revised version of the manuscript that addresses the points raised during the review process.

We look forward to receiving your revised manuscript.

Kind regards,

Yu Zhou

Academic Editor

PLOS ONE

Journal Requirements:

"Supported by Ministry of Education, Industry-University Cooperation Collaborative Education 

Project (230825052507181). 

Funded by Science Research Project of Hebei Education Department (BJS2024097). 

Supported by Hebei Province Social Science Development Research Project (20230303051)."

"Supported by Ministry of Education, Industry-University Cooperation 

Collaborative Education Project (230825052507181). Funded by Science Research Project of Hebei Education Department (BJS2024097). Supported by Hebei Province Social Science

 Development Research Project (20230303051)."

"Supported by Ministry of Education, Industry-University Cooperation Collaborative Education 

Project (230825052507181). 

Funded by Science Research Project of Hebei Education Department (BJS2024097). 

Supported by Hebei Province Social Science Development Research Project (20230303051)."

5. We note that your Data Availability Statement is currently as follows: [All relevant data are within the manuscript and its Supporting Information files.]

6. Please upload a copy of Figure 8, to which you refer in your text on page 14. If the figure is no longer to be included as part of the submission please remove all reference to it within the text.

7. We note that [Figures 2-4] in your submission contain [map/satellite] images which may be copyrighted. All PLOS content is published under the Creative Commons Attribution License (CC BY 4.0), which means that the manuscript, images, and Supporting Information files will be freely available online, and any third party is permitted to access, download, copy, distribute, and use these materials in any way, even commercially, with proper attribution. For these reasons, we cannot publish previously copyrighted maps or satellite images created using proprietary data, such as Google software (Google Maps, Street View, and Earth). For more information, see our copyright guidelines: http://journals.plos.org/plosone/s/licenses-and-copyright.

a. You may seek permission from the original copyright holder of Figures 2-4 to publish the content specifically under the CC BY 4.0 license.  

Reviewers' comments:

Reviewer's Responses to Questions

**Comments to the Author**

1. Is the manuscript technically sound, and do the data support the conclusions?

Reviewer #1: Partly

Reviewer #2: Partly

Reviewer #3: Yes

2. Has the statistical analysis been performed appropriately and rigorously? 

Reviewer #1: No

Reviewer #2: I Don't Know

Reviewer #3: Yes

3. Have the authors made all data underlying the findings in their manuscript fully available?

Reviewer #1: No

Reviewer #2: Yes

Reviewer #3: Yes

4. Is the manuscript presented in an intelligible fashion and written in standard English?

Reviewer #1: Yes

Reviewer #2: Yes

Reviewer #3: Yes

5. Review Comments to the Author

Reviewer #1: 1.The title does not provide a core theme of the topic.

2.Please specify the source of the simulation data.

3.The language of this manuscript needs help from native speakers.

4.Please underscore the scientific value-added to your paper in your abstract. Your abstract should clearly state the essence of the problem you are addressing, what you did and what you found and recommend. That would help a prospective reader of the abstract to decide if they wish to read the entire article. Abstract: Please underscore the scientific value-added to your paper in your abstract. It might be helpful to provide a summary at the beginning of both the abstract and conclusions to give readers a clear overview of the main findings and implications.

5.Introduction: Clearly articulate the objectives of the study and hypotheses at the beginning of the paper. Readers should be aware of what the study intends to achieve and what the predicted outcomes. it is recommended that the author refer to the paper on Does digital inclusive finance promote the integration of rural industries? Based on the mediating role of financial availability and agricultural digitization. or similar topics for modification

6.Suggest redrawing the research framework for readers to better understand.

7.In the model of section Spatial econometric analysis of this article, it is recommended that the author refer to the paper on Does environmental pollution governance contribute to carbon emission reduction under heterogeneous green technological innovation?. or similar topics for modification.

8.Elaborate more on the source and type of data used, and ensure the reliability and validity of the data are addressed.

9.There are a lot of grammatical mistakes and spelling mistakes, please check and polish the paper totally..

Reviewer #2: Under the new development pattern, the main driving force for promoting the highquality development of agriculture and rural areas is the development of "digitalization" and " low-carbonization". This is also the key task and important guarantee for the comprehensive implementation of the strategy of rural revitalization. Using provincial panel data from 2013 to 2021, this paper employs the entropy value method, SBM model, and coupling coordination degree model to investigate the coupling coordination measurement and spatial-temporal heterogeneity of the performance of the digital economy and agricultural carbon emissions.

Overall, I think this manuscript is suitable for this journal’s scope. However, there are some issues that may need to be improved.

Abstract. The description of the study context/importance of the study is relatively good, the study methods and data are clearly described. However, if possible，the findings part should be more accurate and detailedly. The two parts of method use and background meaning can be simplified a little.

Introduction. The novelty of this paper should be further justified by highlighting main contributions to the existing introduction and literature review. For example, what are the other researches on the whole CO2 emission in China? E.g., one entitled “China's CO2 emissions: A systematical decomposition concurrently from multi-sectors and multi-stages since 1980 by an extended logarithmic mean divisia index. https://doi.org/10.1016/j.esr.2023.101141”.

Theoretical analysis or Literature review. Here，it is suggested that the author should strengthen the extraction and summary. The relevant literature should also receive attention. For example，e.g.，one entitled “China's CO2 Emissions: A Thorough Analysis of Spatiotemporal Characteristics and Sustainable Policy from the Agricultural Land-Use Perspective during 1995-2020. https://doi.org/10.3390/land12061220”.

Equation. There is something wrong with the order in which the formulas are written.

Tables. There is something wrong with the pattern in which the tables are written. E.g, in the case of table 9, this style is problematic..

Figures. The text in the picture is not clear.

In the whole text， the YUAN or RMB should be CNY. The full text should be carefully proofread. Many typographical and editing errors.

Grammar. The writer should proofread the whole text carefully.

References, the author should carefully check all the references, all references should be formatted in accordance with the standards of the journal

Reviewer #3: The overall analysis of this article is quite rigorous and has research significance.

The following details should be noted for revision: 1. The introduction section is too lengthy and lacks conciseness when introducing previous research. It is recommended to fill it out again.

The Limitations of this study section can be abbreviated.

3. The language of the entire article needs to be appropriately polished to make it easier to understand.

6. PLOS authors have the option to publish the peer review history of their article (what does this mean?). If published, this will include your full peer review and any attached files.

Reviewer #1: No

Reviewer #2: No

Reviewer #3: No

---

## [Author Response · Author response to Decision Letter 0]

1 May 2024

Dear Paula Katrina A. Maderazo：

I am gratified to receive your email and would like to offer the following clarification in response to your reference to the map in the manuscript.

Figures 2, 3, and 4 in the manuscript were created with copyright in mind. The base map, shapefile, or map image used to create these figures was obtained from Natural Earth (http://www.naturalearthdata.com/). Concurrently, the manuscript's legend section is incorporated with a corresponding note.

Note：Natural Earth was built through a collaboration of many volunteers and is supported by NACIS (North American Cartographic Information Society), and is free for use in any type of project .

Yours sincerely

Wu Yuhuan

May 1, 2024

Department of Economics and Trade,

Hebei University of Water Resources and Electric Engineering,Cangzhou , Hebei

P.R.China

Tel: 86-0317-7587114

Email: wuyuhuan@hbwe.edu.cn

---

## [Decision Letter · Decision Letter 1]

28 May 2024

Measurement and Spatio-temporal Heterogeneity Analysis of Coupling Coordination between development of Digital Economy and Agricultural Carbon Emission Performance

PONE-D-24-07374R1

Dear Dr. Wu,

We’re pleased to inform you that your manuscript has been judged scientifically suitable for publication and will be formally accepted for publication once it meets all outstanding technical requirements.

Kind regards,

Yu Zhou

Academic Editor

PLOS ONE

Additional Editor Comments (optional):

Reviewers' comments:

Reviewer's Responses to Questions

**Comments to the Author**

1. If the authors have adequately addressed your comments raised in a previous round of review and you feel that this manuscript is now acceptable for publication, you may indicate that here to bypass the “Comments to the Author” section, enter your conflict of interest statement in the “Confidential to Editor” section, and submit your "Accept" recommendation.

Reviewer #2: All comments have been addressed

Reviewer #4: (No Response)

2. Is the manuscript technically sound, and do the data support the conclusions?

Reviewer #2: Yes

Reviewer #4: (No Response)

3. Has the statistical analysis been performed appropriately and rigorously? 

Reviewer #2: Yes

Reviewer #4: (No Response)

4. Have the authors made all data underlying the findings in their manuscript fully available?

Reviewer #2: No

Reviewer #4: (No Response)

5. Is the manuscript presented in an intelligible fashion and written in standard English?

Reviewer #2: Yes

Reviewer #4: (No Response)

6. Review Comments to the Author

Reviewer #2: I think the quality of the improved article has improved significantly.

I suggest that the article can be published after being polished according to the format requirements of the journal.

Reviewer #4: The manuscript titled "Measurement and Spatio-temporal Heterogeneity Analysis of Coupling Coordination between development of Digital Economy and Agricultural Carbon Emission Performance" holds practical significance for the development of China's digital economy and agricultural carbon emission. There are still some problems in the revised manuscript, which I suggest to accept after modification. The problems are as follows:

1. The discussion section lacks literature support, please add corresponding references.

2.The picture is a little fuzzy, please improve the clarity of the picture.

7. PLOS authors have the option to publish the peer review history of their article (what does this mean?). If published, this will include your full peer review and any attached files.

Reviewer #2: No

Reviewer #4: No

---

## [Editor Report · Acceptance letter]

30 May 2024

PONE-D-24-07374R1 

PLOS ONE

Dear Dr. Wu, 

I'm pleased to inform you that your manuscript has been deemed suitable for publication in PLOS ONE. Congratulations! Your manuscript is now being handed over to our production team.

Kind regards, 

on behalf of

Dr. Yu Zhou 

Academic Editor

PLOS ONE